# Spinal neural tube formation and tail development in human embryos

**Chloe Santos[†], Abigail R Marshall[†], Ailish Murray[†‡], Kate Metcalfe, Priyanka Narayan, Sandra CP de Castro, Eirini Maniou[§], Nicholas DE Greene, Gabriel L Galea, Andrew J Copp***

Developmental Biology & Cancer, UCL Great Ormond Street Institute of Child Health, London, United Kingdom

**\*For correspondence:**
a.copp@ucl.ac.uk

[†]These authors contributed equally to this work

**Present address:** [‡]Moorfields Eye Charity, London, United Kingdom; [§]Department of Industrial Engineering, University of Padua and Veneto Institute of Molecular Medicine, Padua, Italy

**Competing interest:** The authors declare that no competing interests exist.

## eLife Assessment

This is a **fundamental** study into human spinal neurulation, which substantially advances our understanding of human neural tube closure. Crucial unanswered questions in the field currently rely on model systems, not faithful to human development. The evidence provided is **compelling**, with a large number of specimens and the rigorous use of state-of-the-art methodology providing robustness. The work will be of broad interest to developmental biologists, embryologists, and medical professionals working on neural tube defects, and will act as a precious reference resource for future studies.

**Abstract** Primary and secondary neurulation – processes that form the spinal cord – are incompletely understood in humans, largely due to the challenge of accessing neurulation-stage embryos (3–7 weeks post-conception). Here, we describe findings from 108 human embryos, spanning Carnegie stages (CS) 10–18. Primary neurulation is completed at the posterior neuropore with neural plate bending that is similar, but not identical, to the mouse. Secondary neurulation proceeds from CS13 with formation of a single lumen as in mouse, not coalescence of multiple lumens as in chick. There is no evidence of a 'transition zone' from primary to secondary neurulation. Secondary neural tube 'splitting' occurs in 60% of proximal human tail regions. A somite is formed every 7 hr in human, compared with 2 hr in mice and a 5 hr 'segmentation clock' in human organoids. Termination of axial elongation occurs after down-regulation of *WNT3A* and *FGF8* in the CS15 embryonic tailbud, with a 'burst' of apoptosis that may remove neuro-mesodermal progenitors. Hence, the main differences between human and mouse/rat spinal neurulation relate to timing. Investigators are now attempting to recapitulate neurulation events in stem cell-derived organoids, and our results provide 'normative data' for interpretation of such research findings.

## Introduction

Development of the lumbosacral spinal cord is a critical period of embryogenesis. Not only does motor control and sensation in the legs and lower body depend on this event, but proper functioning of bladder, rectum, and genital organs are all critically dependent on nerves arising from the low spinal cord. A major group of congenital malformations termed neural tube defects result when low spinal neurulation fails to be completed or is otherwise abnormal, and these can be open or closed (skin-covered) lesions.

Open spina bifida (also called myelomeningocele) results from defective closure of the primary neural tube, most often at lumbar and upper sacral levels. Subsequent neuroepithelial damage and loss due to prolonged exposure to amniotic fluid (*Stiefel et al., 2007*) leads to a defect that is disabling

in most individuals (*Copp et al., 2015*). Closed 'dysraphic' conditions arise at lower sacral and coccygeal levels of the body axis and result from disturbance of secondary neurulation, in which the neural tube forms without formation of neural folds. Dysraphic conditions involve an abnormal anatomical relationship between the secondary neural tube and surrounding tissues, often with ectopic adipose tissue, as in spinal lipoma and lipomyelomeningocele (*Jones et al., 2019*). Closed dysraphism may be asymptomatic, but significant disability can occur through tethering of the low spinal cord to non-neural tissues (*Agarwalla et al., 2007*).

In normal primary and secondary neurulation, the neural tube forms by closure and canalisation respectively (summarised in Figure 1 of *Nikolopoulou et al., 2017*), with extensive studies of both processes in experimental animals, especially chick, mouse, and rat. Despite the relative inaccessibility of neurulation-stage human embryos (3–7 weeks post-conception), a number of studies have described the anatomical, histological, and ultrastructural features of secondary neurulation (*Supplementary file 1*). Although limited molecular research has been performed, e.g., to determine the mode of cell death in human tail regression (*Vilović et al., 2006*), a few studies have begun to address specific gene expression during human secondary body development (*Olivera-Martinez et al., 2012*; *Yang et al., 2014*). Increasingly, transcriptomic approaches are being used with organogenesis-stage human embryos (*Fang et al., 2010*; *Yi et al., 2010*; *Krupp et al., 2012*; *Xu et al., 2023*).

Caudal development comprises not only formation of the secondary neural tube, but also other tissue types within the 'secondary body' region. This part of the body axis – beyond the cloacal plate which marks the future anus – includes the secondary notochord, tail somites, caudal vessels, tailgut, and surrounding surface ectoderm (future epidermis). These structures show marked tissue-to-tissue variation in development. For example, tail regression in human embryos involves loss of all tail components, whereas the rodent tail maintains the somites and notochord, but loses the secondary neural tube and tailgut. The regulation of this balance between maintenance and loss of tail structures is not understood.

A related area of interest is the molecular control of axial elongation. A population of self-renewing stem cells, termed neuro-mesodermal progenitors (NMPs), resides in the caudal-most embryonic region (the tailbud), with NMPs giving rise to neural and mesodermal derivatives, including the secondary neural tube and somites (*Henrique et al., 2015*). NMP maintenance is required for axial elongation, mediated via an interplay between *WNT3A* and *FGF8* expression, which promotes NMP survival in the tailbud. Conversely, endogenous retinoic acid promotes differentiation and regulates body length (*Wilson et al., 2009*).

In the present study, we examined caudal development in 108 human embryos, at Carnegie stages (CS) 10–18 (3.5–6.5 weeks post-conception). The aim was to gain new information on several unanswered or controversial questions in human neurulation, including: (i) how the embryo transitions from primary to secondary neurulation; (ii) the mode of formation of the secondary neural tube; (iii) the rate of somite formation during low spinal development; (iv) the possible roles of *WNT3A* and *FGF8* in regulating axial elongation; (v) whether a 'burst' of apoptosis coincides with termination of axial elongation. A further aim was to gather and present findings on human spinal neurulation that can serve as 'normative data' to aid interpretation of research involving multicellular 'organoid' structures, that are being increasingly used to model various aspects of human axial development (*Denham et al.,*

**Table 1.** Number of human embryos in the study, with breakdown by analysis type, sex, and method of pregnancy termination*.

| Analysis type | Figures in paper | Total no. | No. females | No. males | No. sex unknown | No. medical | No. surgical |
|---|---|---|---|---|---|---|---|
| PNP morphology | 1A–H | 2** | 0 | 2 | 0 | 2 | 0 |
| PNP closure timing | 1I, J | 40 | 24 | 16 | 0 | 40 | 0 |
| Tail morphology, histology, cell death | 2, 3 | 37** | 14 | 21 | 2 | 36 | 1 |
| Serial section analysis, cell death | 4, 6 | 11 | 5 | 6 | 0 | 10 | 1 |
| *FGF8*, *WNT3A* expression | 5 | 18 | 9 | 9 | 0 | 18 | 0 |
| Totals | | 108 | 52 | 54 | 2 | 106 | 2 |

*Medical: mifepristone- and misoprostol-induced delivery; Surgical: ultrasound-guided vacuum aspiration.
**Embryos that are included in *Table 2*.

*2015*; *Fedorova et al., 2019*; *Moris et al., 2020*; *Rifes et al., 2020*; *Karzbrun et al., 2021*; *Libby et al., 2021*; *Amadei et al., 2022*).

## Results

The study involved 108 human embryos (*Table 1*), obtained from the Medical Research Council (MRC)/Wellcome Human Developmental Biology Resource (https://www.hdbr.org/), with UK ethics committee approval. Embryos were donated by women undergoing termination of pregnancy for 'social' reasons, in most cases by mifepristone/misoprostol-induced (medical) delivery, with a few intact embryos obtained by ultrasound-guided vacuum aspiration (surgical). All embryos in the study were chromosomally and morphologically normal, and were assigned to CS, as described (*O'Rahilly and Muller, 1987*; *Bullen and Wilson, 1997*). Comparisons to mouse were with random-bred CD1 embryos, staged by embryonic (E) day, where E0.5 is the day following overnight mating.

### Morphology of human PNP closure

Relatively few human embryos with an open posterior neuropore (PNP) have been reported in the literature (*Müller and O'Rahilly, 1987*; *O'Rahilly and Müller, 2002*), probably owing to the early stage at which primary neurulation is completed (end of week 4, post-conception). In two intact CS12 embryos (*Figure 1A and B*; crown-rump length: 3 mm; 22–23 somites), we identified an open PNP by microscopic inspection at collection (*Figure 1C and D*). Transverse histological sections confirmed an open neural tube in the caudal region, with minimal tissue damage evident, indicating that primary neurulation was not yet complete. The neural plate is relatively flat in the most caudally located sections, although incipient dorsolateral hinge points (DLHPs) are visible (*Figure 1E and F*). The notochord underlies the neural plate midline, and the caudal end of the hindgut is visible beneath the notochord in one embryo (*Figure 1E*), but not the other (*Figure 1F*). In more rostral sections, close to the 'zippering' point of PNP closure, elevated neural folds flank a marked ventral midline bend in the neural plate, the median hinge point (MHP), which precisely overlies the notochord (*Figure 1G and H*). DLHPs are also clearly present, unilaterally in one embryo (*Figure 1G*) and bilaterally in the other (*Figure 1H*). As in the mouse (*McShane et al., 2015*), the DLHPs are situated where the neural plate changes from basal contact with surface ectoderm to basal contact with paraxial mesoderm. We conclude that MHP and DLHPs characterise PNP closure in human embryos at CS12, marking a direct equivalence to Mode 2 spinal neurulation in the mouse embryo (*Shum and Copp, 1996*).

### Timing of human PNP closure

PNP length data were obtained from photographic images of CS10–13 embryos (n=40). To allow for differences in overall embryonic size, PNP measurements were normalised to the length of a recently formed somite in the same embryo (*Figure 1C*). The plot of PNP length/somite length against somite number shows a steady decline in the length of open neural folds in the caudal region, until 6/12 embryos at CS13 have completely closed, while most of the others show a very small PNP (*Figure 1I*). There were no obvious differences in closure rate or timing between female (n=24) and male (n=16) embryos. Hence, closure of the PNP is completed in human embryos around the 30-somite stage, as also reported for outbred mouse strains (*Copp et al., 1982*).

### Development and regression of the human embryonic tail

Overall caudal development was studied in 37 human embryos (CS13–18), which covered the period 28–45 days post-conception (*Table 2*; *Figure 2A, B, I*). Crown-rump length increased 2.5-fold during this period, from a mean value of 6.4 mm at CS13 to 15.4 mm at CS18 (*Table 2*; *Figure 2J*). Observations on the intact embryos showed that the PNP is closed in most embryos by CS13, and a developing tailbud is present which exhibits mild ventral curvature and a thick rounded tip (*Figure 2C*). Somites are visible proximal to the tailbud (arrowheads in *Figure 2C*), with an intervening region of presomitic mesoderm at CS13 (yellow bracket in *Figure 2C*). By CS16, however, the somites extend almost to the tail tip (yellow arrow in *Figure 2F*). As development progresses, striking changes occur in the tail which continues to lengthen (*Table 2*) but simultaneously narrows, particularly at the tip, to yield a sharply pointed structure by CS16 (*Figure 2D–F*). At the same time, the tail straightens and even becomes dorsally bent (*Figure 2F*). Subsequent to CS16, the tail shortens (*Figure 2G*; *Table 2*),

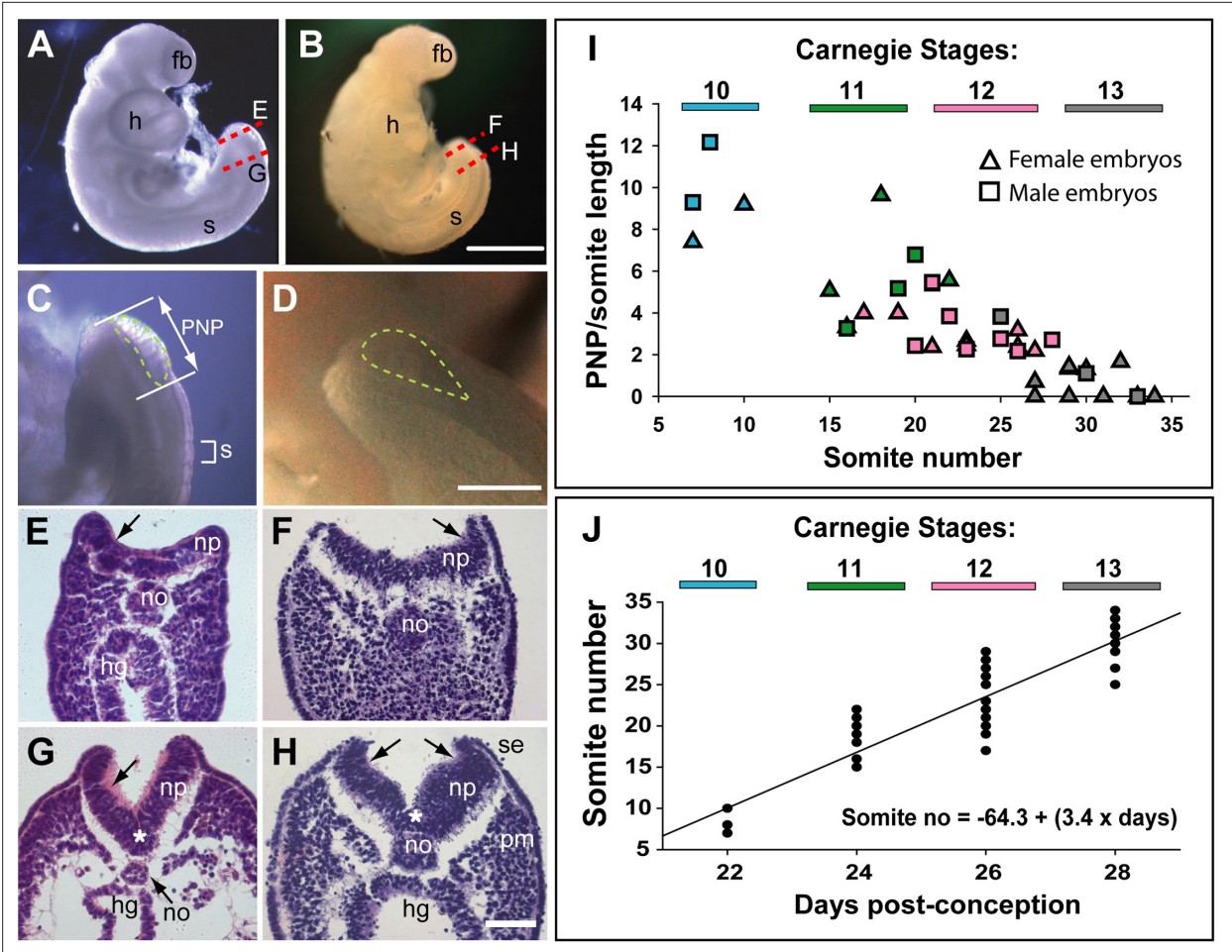

**Figure 1.** Morphology and timing of human posterior neuropore (PNP) closure. (**A,B**) Two CS12 embryos viewed from right side, each with 22–23 somites (**s**), and a looped heart (**h**). Neural tube closure is complete along most of the body axis, including the forebrain (**fb**), whereas the PNP remains open caudally. (**C,D**) Magnified oblique views from upper right side of the caudal region; the open PNP is outlined with dashed lines. (**E–H**) Haematoxylin and eosin (H&E)-stained transverse sections, through the PNP, with section planes as indicated by dashed lines in A,B. The most caudally located sections (**E,F**) show a relatively flat neural plate (np), although incipient dorsolateral hinge points (DLHPs; arrows) are visible. Note the midline notochord (no) underlying the neural plate, and hindgut (hg) beneath the notochord (in E only). More rostral sections (**G,H**) show elevated neural folds with DLHPs clearly visible (arrows: unilateral in G, bilateral in H), located where basal contact of the neural plate changes from surface ectoderm (se), to paraxial mesoderm (pm). A median hinge point (MHP; asterisks in G,H) overlies the notochord. (**I**) PNP length (double headed arrow in C), normalised to somite (**s**) length (bracketed in C), determined from photographic images of 40 human embryos (24 females; 16 males) at CS10 (n=4), CS11 (n=8), CS12 (n=16), and CS13 (n=12). Symbol colours indicate the Carnegie stages assigned at the time of collection. The PNP shows gradual closure, with completion around the 30 somite stage. (**J**) Somite number of the 40 embryos in I, plotted against days post-conception, as reported for each Carnegie stage by *O'Rahilly and Muller, 1987*. The linear regression equation is shown, with R²=0.82 and p<0.001. Scale bars: 1 mm in A,B; 0.4 mm in C,D; 0.1 mm in E–H.

The online version of this article includes the following source data for figure 1:

**Source data 1.** Measurements from photographic images of individual human embryos, as described in Materials and methods.

---

and its distal portion becomes increasingly translucent in appearance. By CS18, only a short, curved stump remains (*Figure 2H*), and the tail is lost completely thereafter. Additional embryonic tails in the CS13–18 range are shown in *Figure 2—figure supplement 1*.

## Somite formation

Between CS10 and CS13, during PNP closure, somite number increases approximately linearly with days of gestation (*Figure 1K*): mean (± SD) somite numbers were: 8.0±1.4 at CS10, 18.4±2.6 at CS11, 23.5±3.4 at CS12, and 30.0±2.8 at CS13. The linear regression equation of this relationship (*Figure 1K*) gives an increase of 20.3 somites over a 6-day period, equating to the formation of 3.4

---

**Table 2.** Measurements of human embryos, CS12–18*.

| Carnegie stage (CS) | Age range (days post-fertilisation) | Number of embryos | Somite number [†] | Crown-rump length [‡] | Tail length (total) [‡] | Tail length distal to somites [‡] |
|---|---|---|---|---|---|---|
| 12 | 25–27 | 2 | 22, 22 | 3.0, 3.0 | N/A | N/A |
| 13 | 28–30 | 7 | 35.4±2.3 | 6.4±1.3 | 1.06±0.44 | 0.49±0.16 |
| 14 | 31–32 | 5 | 35.0±2.0 | 8.4±2.2 | 0.99±0.21 | 0.56±0.06 |
| 15 | 33–35 | 7 | 35.9±1.8 | 8.9±0.8 | 1.18±0.55 | 0.67±0.15 |
| 16 | 37–39 | 8 | 36.6±2.6 | 11.7±1.2 | 1.29±0.48 | 0.46±0.13 |
| 17 | 40–43 | 4 | 32.3±2.1 | 12.2±1.2 | 1.18±0.20 | 0.34±0.01 |
| 18 | 44–45 | 6 | 32.6±1.1 | 15.4±2.2 | 1.14±0.48 | N/D |

N/A: not applicable; N/D: not determined.

*Somite numbers: mean ± SD (except CS12, where actual somite numbers are shown). Somite number was available for all embryos except n=5 at CS18.

[†]Summary of embryos that underpin **Figure 1A–H** (CS12) and **Figures 2 and 3** (CS13–18). For full data set, see **Supplementary file 2**.

[‡]Lengths (mm): mean ± SD (except CS12, where actual lengths are shown). Length measuements were available only for a subset of embryos. See **Supplementary file 2** for full details.

somites per gestational day, or a new somite every 7.1 hr (95% confidence intervals: 4.8, 10.4). This compares with formation of a new somite every 2 hr in rat and mouse embryos (**Brown and Fabro, 1981**; **Tam, 1981**), and a 5 hr periodicity observed for the human 'in vitro segmentation clock' in stem cell-derived presomitic mesoderm-like cells (**Diaz-Cuadros et al., 2020**; **Matsuda et al., 2020**). Following PNP closure at CS13, the largest somite number was at CS16 (36.6±1.2; **Table 2**), although there was no statistically significant increase between CS13 and CS16 (**Figure 2K**). By CS17 and 18, we could identify only 31–34 somites, a significant reduction in number (**Figure 2K**). Hence, somite formation in humans occurs at a rate that is 3.5 times slower than in rodent embryos, and ceases after CS16. Subsequent somite number reduction suggests that shortening of the tail during regression involves loss of somites (**Table 2**).

## Mode of cell death during tail regression

In an initial study, transverse histological sections through human and mouse embryonic tails were processed for immuno-peroxidase staining using anti-activated caspase 3. Positive cells were readily identified in the tails of both species (**Figure 3A–N**), arguing for a role of caspase-dependent apoptosis during tail regression in human and mouse. Principal sites of apoptotic cell death include the regressing tailgut (**Figure 3D and F**) and, most abundantly, the ventral mesoderm overlying the epithelial ventral ectodermal ridge (**Figure 3C and E**). We also detected terminal deoxynucleotidyl transferase dUTP nick end labelling (TUNEL)-positive cells in both mouse and human tail sections (data not shown), further confirming the presence of apoptotic cells during tail development/regression.

## A burst of apoptosis at cessation of tail elongation

We observed enhanced apoptosis in the mouse tailbud at E13.5 (**Figure 3J**), compared with E13.0 and E14.0 when relatively few dying cells were present (**Figure 3I and K**). Similarly, in sections through the caudal-most region of human embryos, apoptosis was not observed at CS13 (**Figure 3L**), became intense at CS15 (**Figure 3M**), and diminished in intensity by CS18 (**Figure 3N**). Hence, in both mouse and human tails, there appears to be a 'burst' of apoptosis at the stage when tail growth ceases, and just before regression of internal structures gets underway.

## Evidence for regression of the tailgut from rostral to caudal

While the human embryonic tail appears to regress from caudal to rostral (**Figure 2**), the proximal (rostral) part of the tailgut is reported to degenerate before the more distal (caudal) part in both rat (**Butcher, 1929**; **Qi et al., 2000**) and mouse (**Nievelstein et al., 1993**). To examine this question in human embryos, we performed immunofluorescence for anti-activated caspase 3, which confirmed

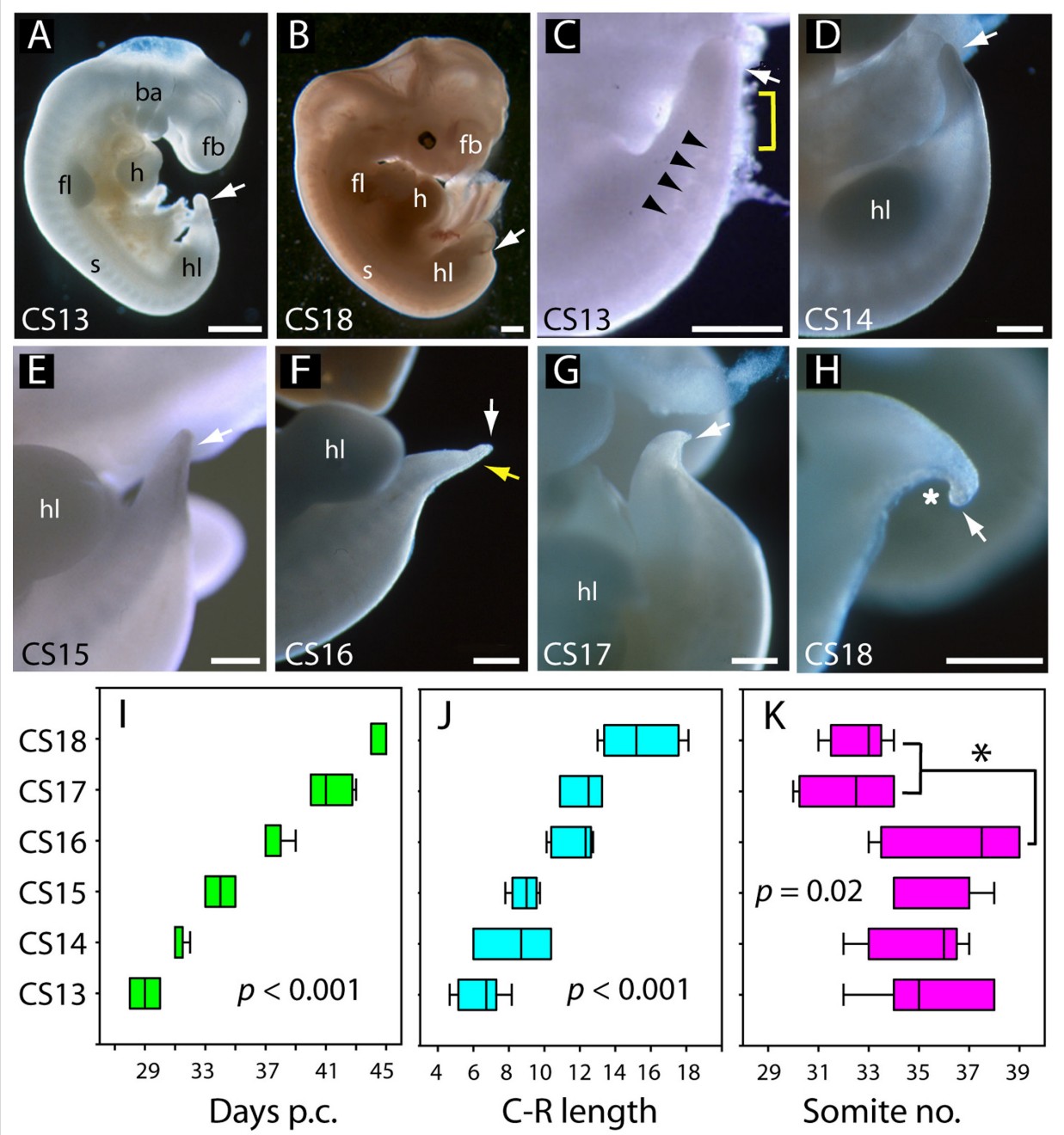

**Figure 2.** Development of the tail in human embryos. (**A,B**) Whole embryos at CS13 (**A**) and CS18 (**B**), showing the range of stages studied (4–6.5 weeks post-conception). The tailbud (arrow) is well formed at CS13 following completion of posterior neuropore (PNP) closure at CS12, whereas, by CS18, tail development and regression are largely complete and only a small tail remnant remains (arrow). (**C–H**) Higher magnification views of the caudal region at CS13–18. At CS13, the tailbud is relatively massive, tapering gradually and with a rounded end (arrow in C). Somites are visible rostral to the tailbud (arrowheads) with an intervening region of presomitic mesoderm (yellow bracket). At CS14 and CS15 the tail narrows progressively, with distal tapering (arrows in D–E). By CS16, this has yielded a slender structure with a narrow pointed end (white arrow in F) in which somites extend almost to the tail tip (yellow arrow in F). Thereafter, the tail shortens progressively (arrows in G,H), develops a marked flexion (asterisk in H), and becomes increasingly translucent (**G, H**). (**I–K**) Analysis of embryos in the range CS13–16 (***Table 2***), plotting CS against: (**I**) days post-conception (p.c., see Materials and methods), (**J**) crown-rump (C–R) length in mm, and (**K**) somite no. One-way analysis of variance (ANOVA) on ranks shows all three parameters vary significantly with CS (p-values on graphs). Somite no. reduces significantly between CS16 and CS17/18 (*$p<0.05$). Abbreviations: ba, branchial arches; fb, forebrain; fl, forelimb; h, heart; hl, hindlimb; s, somites. Scale bars: 1 mm in A,B; 0.5 mm in C–H.

The online version of this article includes the following figure supplement(s) for figure 2:

**Figure supplement 1.** Human embryonic tails (additional to those in ***Figure 2***) to show the reproducibility of tail morphology changes between CS13 and CS18.

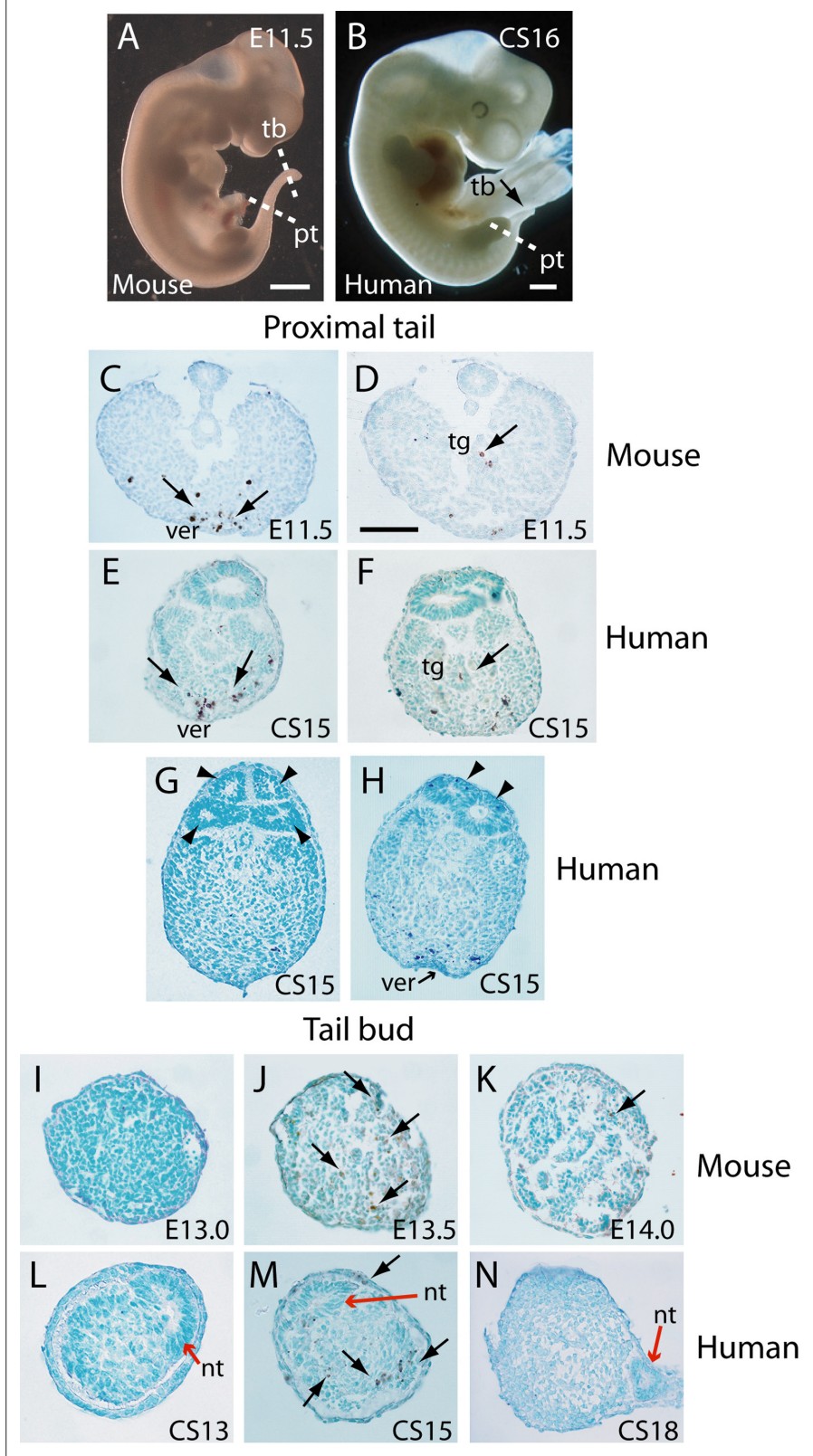

**Figure 3.** Tail morphology and apoptosis in mouse and human embryos. (**A,B**) Mouse E11.5 (**A**) and human CS16 (**B**) embryos to illustrate the level of transverse sections through the proximal tail (pt) and tailbud (tb) regions of mouse (**C,D,I–K**) and human (**E–H,L–N**) embryos at the stages indicated on the panels. Immunohistochemistry was performed on paraffin wax sections for activated caspase 3 (brown stain), with counterstaining by methyl green.

*Figure 3 continued on next page*

*Figure 3 continued*

(**C–F**) In the proximal tail region, intense programmed cell death is observed in the ventral midline mesoderm overlying the ventral ectodermal ridge (ver) of both mouse (arrows in C) and human (arrows in E) embryos. Cell death can also be detected in the tailgut (tg) of both mouse (arrow in D) and human embryos (arrow in F). (**C,D**) are sections from a single E11.5 mouse embryo; (E,F) are sections from a single CS15 human embryo. (**G,H**) Multiple neural tube profiles in two human embryonic tails at CS15: four lumens are visible in one embryo (arrowheads in G) and two lumens in a second (arrowheads in H). (**I–K**) In mouse, the tailbud displays a stage-dependent burst of apoptotic cell death at E13.5 (arrows in J), with absence of caspase 3-positive cells 12 hr earlier, at E13.0 (**I**), and only occasional dying cells 12 hr later, at E14.0 (arrow in K). Note the absence of a neural tube at the mouse tailbud tip, and the sparse nature of the tailbud mesenchyme at E14.0. (**L–N**) Human embryonic tailbuds show a similar developmental sequence to the mouse, with absence of cell death at CS13 (**L**), abundant dying cells at CS15 (arrows in M) and cessation of cell death by CS18 (**N**). Unlike the mouse, the secondary neural tube extends to the tailbud tip (red arrows in L–N), and this terminal neural tube portion has a single lumen in all three embryos. Scale bars in A,B, 1 mm; bar in C represents: 70 µm (**C,D**), 50 µm (**E–K,N**), and 30 µm (**L,M**).

the presence of apoptosis in both tailgut and ventral mesoderm (*Figure 4A' and B'*). Using DAPI (4', 6-diamidino-2-phenylindole)-stained sections along the secondary body axis (*Figure 4A and B*), we determined the cross-sectional area of neural tube, notochord, and tailgut in two CS14 and one CS15 embryos. Total tail area served as a measure of axial position. The neural tube showed a progressive increase in area towards the proximal (rostral) end of the tail, while the notochord showed no change in area along the body axis (*Figure 4C*). Strikingly, the tailgut showed the reverse trend, with a marked caudal-to-rostral reduction in cross-sectional area (*Figure 4A, B, and D*). Tailgut nuclear number also diminished from caudal to rostral (*Figure 4E*). Hence, although the rostral tailgut has not disappeared by CS15, it appears to be diminishing in size proximally, at the same time as it is being formed, as a prominent tail structure, caudally. We conclude that rostral-to-caudal loss of the tailgut may be a general phenomenon among mammalian embryos.

## Expression of *FGF8* and *WNT3A* during human tailbud elongation

To begin an assessment of the mechanisms that may regulate elongation of the human embryonic tail, and its cessation, we performed whole-mount in situ hybridisation for *FGF8* and *WNT3A* (n=2 embryos minimum for each gene at each stage). These genes are developmentally regulated during axial elongation in chick and mouse embryos, with strong expression in the tailbud during elongation, and down-regulation before axial growth ceases. Direct inactivation or indirect down-regulation of the genes leads to premature axial truncation (*Wilson et al., 2009*).

In accordance with these findings, we observed strong expression of *FGF8* in the tailbud at CS12 and CS13, as revealed in whole embryos (*Figure 5A and B*) and longitudinal sections through hybridised caudal regions (*Figure 5E and F*). At CS14, *FGF8* expression reduced dramatically so that only a small 'dot' of expression was detected in the tailbud (*Figure 5C and G*), and by CS15 expression of *FGF8* was no longer detectable in the tail (*Figure 5D and H*). Expression of *WNT3A* followed a similar pattern with strong expression in the tailbud at CS12 (*Figure 5I and M*), reduced expression intensity at CS13 (*Figure 5J and N*), a remaining 'dot' of tailbud expression at CS14 (*Figure 5K and O*), and no detectable *WNT3A* expression in the tail at CS15 (*Figure 5L and P*). We conclude that expression of *FGF8* and *WNT3A* mirrors the relationship seen in mouse and chick, with strong tailbud expression during active axial extension, and dramatic down-regulation of both genes before the onset of axial growth cessation. It is striking that down-regulation appears complete by CS15, even though the embryonic tail does not reach its maximum length until some days later, at CS16 (*Table 2*). Down-regulation of *Fgf8* and *Wnt3a*, well in advance of cessation of axial elongation, has also been observed in mouse embryos (*Cambray and Wilson, 2007*).

## Mode of secondary neural tube formation in human embryos

In our initial study of tail morphology, 9 out of 15 human embryonic tails showed multiple lumens in some transverse sections (*Figure 3G and H*), whereas the other 6 exhibited only a single neural tube lumen. In a second group of serially sectioned tails (*Figures 4 and 6*), 6 out of 10 tails had regions of duplicated neural tube. Hence, we find a 60% (15/25) frequency of neural tube duplication, confirming previous findings of multiple neural tube lumens in many human embryonic tails (*Bolli, 1966*; *Lemire,*

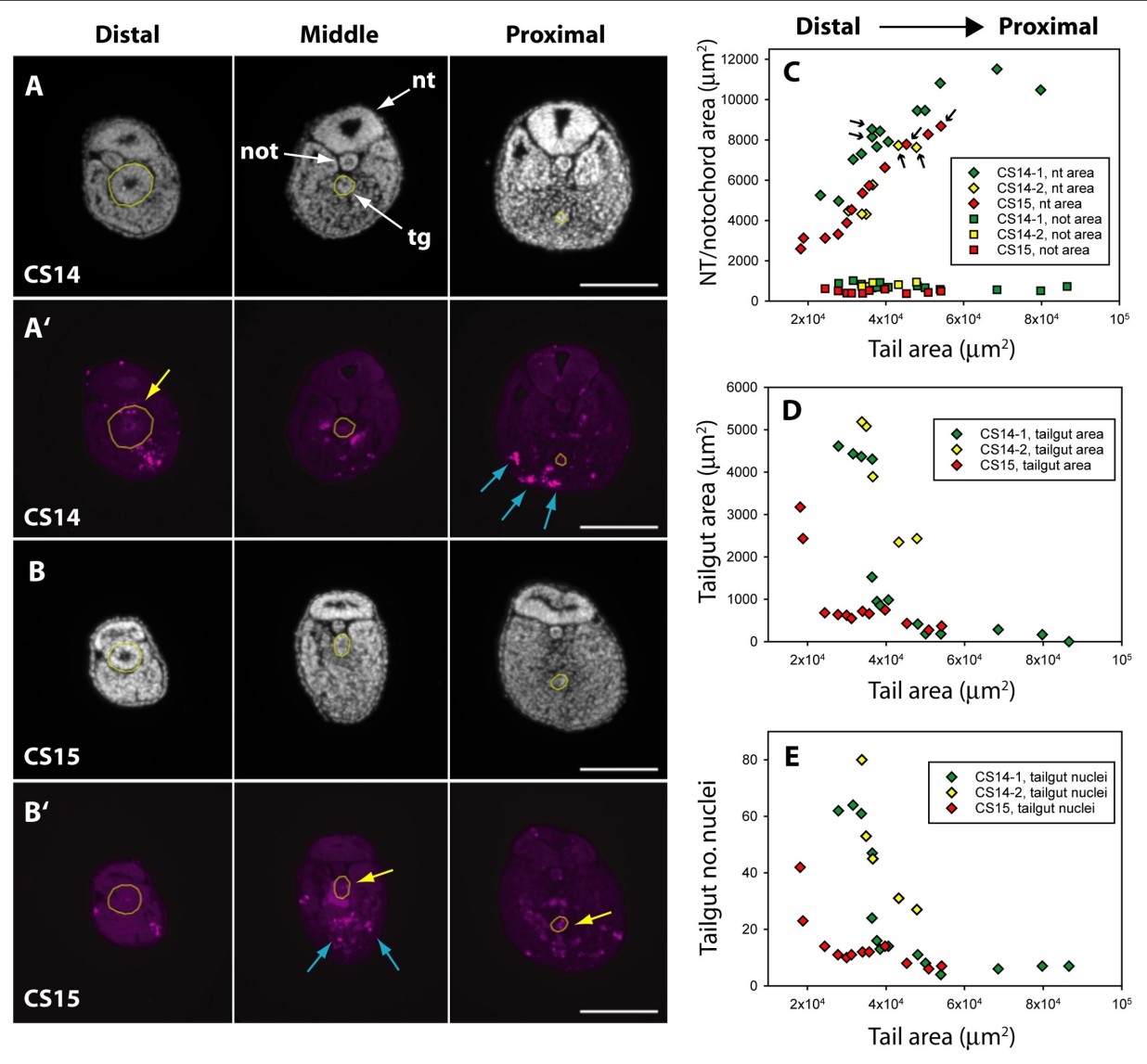

**Figure 4.** Programmed cell death and tissue size along the developing human tail. (**A,A′,B,B′**) Transverse sections at distal (left), middle (centre), and proximal (right) levels of the tail. Panels show DAPI (4′,6-diamidino-2-phenylindole) (**A,B**) and anti-cleaved caspase 3 immunostaining (**A′,B′**) of the same sections at CS14 (**A,A′**) and CS15 (**B,B′**). Yellow dotted lines outline the tailgut. Apoptotic cell death occurs mainly in tailgut (tg, yellow arrows) and ventral mesoderm (blue arrows). Note the diminishing diameter of the tailgut from distal to proximal. (**C**) Change in transverse sectional areas of neural tube (nt, diamonds) and notochord (not, squares) along the body axis in CS14 (x2; green and yellow symbols) and CS15 (red symbols) embryos. Embryos CS14-1 and CS15 are shown in (**A,B**). Tissue-specific areas (y-axis) are plotted against total tail area (x-axis), which increases from left (distal sections) to right (proximal sections). In all embryos, neural tube area increases in a proximal direction, whereas notochord area is relatively constant along the axis. Arrows: sections in which neural tube shows multiple lumens (see **Figure 6**). (**D,E**) Similar analysis for tailgut area (**D**) and tailgut nuclear number (**E**). Both show a dramatic reduction in a distal-to-proximal direction, in contrast to neural tube and notochord. Scale bars: 50 μm.

The online version of this article includes the following source data for figure 4:

**Source data 1.** Quantification of total tail, neural tube (NT), notochord, and tailgut areas (in square microns), and tailgut nuclear counts in three human embryos at CS14 (x2) and CS15.

*1969*; *Saitsu et al., 2004*). We most often identified two neural tube profiles in a single transverse section, but in some cases more were observed (e.g. at CS15; *Figure 3G*).

These initial findings of neural tube duplication prompted a more detailed examination of the suggestion that multiple neural tube lumens represent a mode of human secondary neurulation similar to that seen in the avian embryo (*Lemire, 1969*; *Saitsu et al., 2004*; *Pang, 2020*). In chick, multiple lumens form distally in the tailbud, and coalesce more rostrally to form the secondary neural

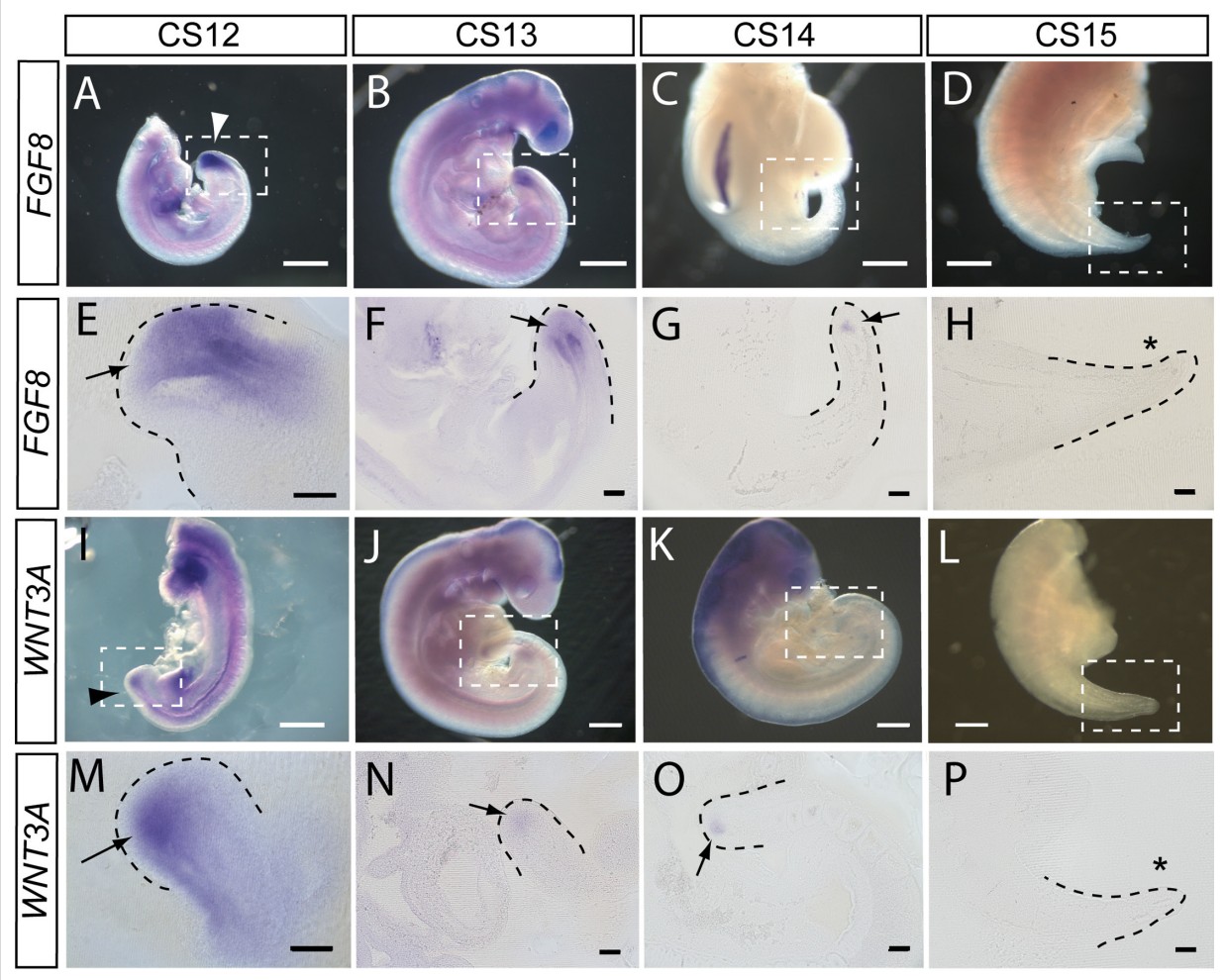

**Figure 5.** *FGF8* and *WNT3A* expression in the elongating caudal region of human embryos. Whole-mount in situ hybridisation (**A–D,I–L**) and sagittal vibratome sections through the caudal region (**E–H,M–P**) for *FGF8* (**A–H**) and *WNT3A* (**I–P**) in embryos at CS12 (**A,E,M**), CS13 (**B,F,J,N**), CS14 (**C,G,K,O**), and CS15 (**D,H,L,P**). Both genes show prominent expression domains in the tailbud at CS12 (arrows in E,M) when axial elongation is underway and the posterior neuropore (PNP) is closing (arrowheads in A,I). At CS13, following PNP closure, expression of *FGF8* and *WNT3A* remains prominent although less intense and more localised to the terminal tailbud than at CS12 (arrows in F,N). By CS14, both genes exhibit much smaller, highly localised expression domains that each appears as a 'dot' within the tailbud region (arrows in G,O). By CS15, axial elongation has ceased, the tail tip has narrowed and is increasingly transparent. At this stage, expression of neither gene can be detected (asterisks in H,P). Whole embryos shown in B,J,K; isolated trunk/caudal regions shown in A,C,D,I,L. No. embryos analysed: FGF8, n=2 for each stage; WNT3A, n=2 for each stage except n=3 for CS13. Scale bars: A–D, I–L, 1 mm; E–H, M–P, 100 μm.

tube (*Criley, 1969*; *Schoenwolf and Delongo, 1980*; *Yang et al., 2003*; *Gonzalez-Gobartt et al., 2021*). An alternative view would be that multiple lumens arise only later in human secondary body formation (*Catala, 2021*), perhaps representing splitting of the already formed secondary neural tube (*Figure 6A*).

To help resolve this issue, we asked two questions: (i) what is the status of the secondary neural tube where it is first formed, close to the tailbud tip? (ii) at what level of the tail axis is neural tube duplication most often seen? In our initial series of human tails, a single lumen was invariably noted in the distal-most portion of the neural tube, just rostral to the tailbud tip (CS13–18; *Figure 3L–N*). This was confirmed in the second series of 10 serially sectioned tails (CS14–15; *Figure 6B, C, and D*), thus demonstrating that secondary neurulation in human produces a neural tube with a single lumen. When multiple secondary neural tube lumens were visible in the embryonic tail, this was invariably located at more rostral levels of the tail (*Figure 3G and H*; *Figure 4C*; *Figure 6B', C", and D"*), with variation in location between embryos. We conclude that multiple secondary neural tube lumens in human embryonic tails likely represent splitting, which tends to occur rostrally and does not represent

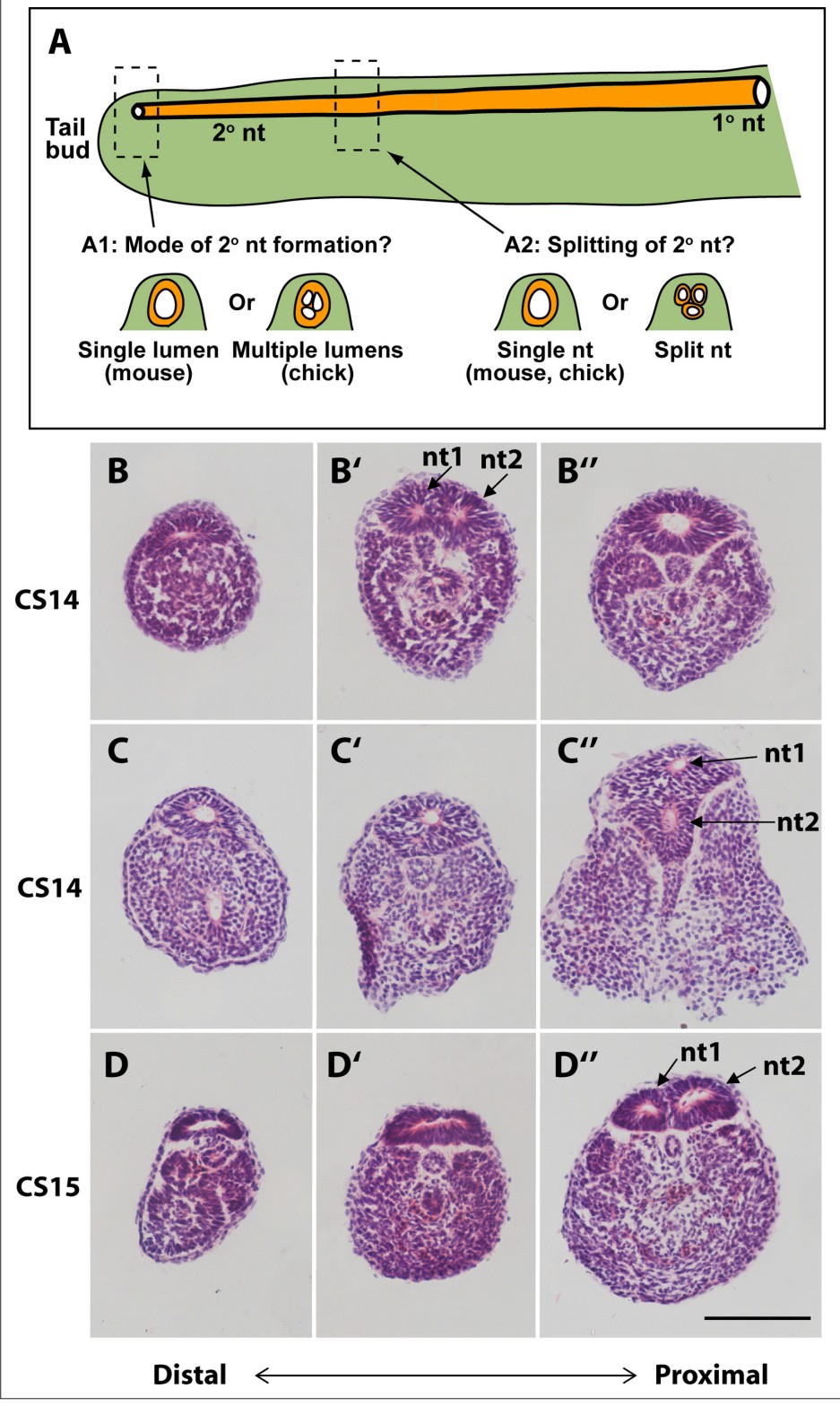

**Figure 6.** Mode of formation and proximal splitting of the human secondary neural tube. (**A**) Hypotheses on mode of formation of the human secondary neural tube (2° nt). A single lumen may be formed in the tailbud as in mouse or, by analogy to chick, multiple lumens may form initially, which then coalesce to form the secondary neural tube (**A1**). Alternatively, the finding of multiple neural tube lumens in sections of some human tails may reflect splitting of the secondary neural tube at more rostral levels (**A2**). (**B–D″**) Representative serial transverse

*Figure 6 continued on next page*

*Figure 6 continued*

sections (haematoxylin and eosin) through three human embryonic tails at CS14 (x2; **B,C**) and CS15 (**D**). Sections close to the tailbud tip (left side: B,C,D) show a broad, dorsoventrally flattened neural tube with a single lumen. There is no evidence of multiple lumens coalescing in the tailbud of any embryos. Further rostrally (middle and right side panels: B',B",C',C",D',D"), some sections show a neural tube with single lumen (**C',D'**), whereas others show evidence of secondary neural tube splitting, with two lumens (nt1, nt2 in B',C",D"). One CS14 embryo shows re-establishment of a single lumen in more proximal sections (**B"**), after splitting more distally (**B'**). These findings support a mouse-like formation of the human secondary neural tube with, additionally, splitting at various rostro-caudal levels along the tail. Scale bar: 50 μm.

a secondary neurulation mechanism involving coalescence of multiple lumens, as in chick. Humans therefore resemble other mammals, including rat and mouse, in initially forming a single secondary neural tube lumen in the tailbud.

## Discussion

The development and later disappearance of the human tail has been of interest to embryologists for more than a century (*Catala, 2021*). To better appreciate the knowledge base for human embryonic caudal development, we conducted a systematic literature review using several search terms (see Materials and methods). This generated a long list of publications that was filtered to include only those with primary data relating to caudal development in human embryos. The final list comprises 28 papers (*Supplementary file 1*) that span over 100 years of research, from the early 1900s (*Kunitomo, 1918*; *Streeter, 1919*) to recent times (*Xu et al., 2023*). This work was based on at least 925 human embryos obtained from varying sources (*Supplementary file 1*) including induced termination of pregnancy for 'social' reasons (where most embryos are expected to be normal), as well as sponta-neous abortion (miscarriage) and ectopic pregnancy, where embryonic abnormalities are likely to be frequent. Hence, interpretation of embryo morphology in these studies needs to take into account the mode of procurement of the human specimens.

Several aspects of caudal development are addressed by the studies in *Supplementary file 1*. These include: completion of primary neurulation with PNP closure, the transition into secondary neurulation, the mode of formation and regression of the secondary neural tube, the observation of multiple neural tube lumens, the formation and regression of somites, notochord and gut in the tail region, the role of programmed cell death in tail regression, and initial studies of gene expression in human embryos. Taken together with the findings of the present study, this accumulated literature provides a strong morphological evidence base for in vivo human caudal development, against which in vitro studies can be judged in the emerging field of stem cell-derived organoid differentiation. The latter is producing multicellular structures that may ultimately provide experimentally tractable models for some aspects of human axial development (*Denham et al., 2015*; *Fedorova et al., 2019*; *Moris et al., 2020*; *Rifes et al., 2020*; *Karzbrun et al., 2021*; *Libby et al., 2021*; *Amadei et al., 2022*).

### Concept of a human 'tail'

The human tail develops and then regresses during weeks 4–7 post-conception and is composed of a secondary neural tube, notochord, somites, and tailgut, with undifferentiated mesenchyme at its tip (the tailbud or 'caudal eminence'), all within a surface ectoderm covering. Since it never becomes vertebrated, in contrast to the tails of most other mammals, some authors consider the human caudal appendage does not qualify as a 'tail' (*Müller and O'Rahilly, 2004*). On the other hand, the presence of caudal somites with vertebra-forming potential are considered by other authors to endow the human caudal appendage with all the hallmarks of a mammalian tail (*Kunitomo, 1918*). In keeping with common usage, we have referred to the transient human caudal appendage as a tail in this paper. Moreover, we note that the recent claim of 'tail-loss' during human evolution, due to a genetic change in the *TBXT* gene (*Xia et al., 2024*), ignores the fact that tail development persists in humans, albeit during a specific stage of embryonic development.

## Transition from primary to secondary neurulation

Primary neurulation ends with completion of PNP closure, which we detect at the 30-somite stage in humans, similar to mice (*Straaten et al., 1992*). In mouse embryos, three sequential 'modes' of spinal neurulation occur (*Shum and Copp, 1996*): in Mode 1, the upper spinal neural plate bends only at the midline (the MHP); in Mode 2, at intermediate spinal levels, bending is at both MHP and paired DLHPs; in Mode 3, immediately before final PNP closure, the MHP is lost and bending occurs only at DLHPs. We found that PNP closure is complete in most CS13 human embryos, and that two CS12 embryos exhibit both MHP and DLHPs in their PNPs. This raises the possibility that 'Mode 2' neural plate bending is typical of the late stages of human spinal neurulation, and that 'Mode 3' is mouse-specific and not seen in humans. Alternatively, our CS12 embryos may have been developmentally too early to show Mode 3 closure. Analysis of further human embryos with open PNPs should resolve this question.

Our sectioning analysis along the body axis of CS14–15 human embryos revealed continuity of the neural tube from proximal (primary neurulation) to distal (secondary neurulation) levels. Indeed, it was not possible to locate with any certainty where the transition from primary to secondary neural tube had occurred. This closely resembles the mouse (*Shum and Copp, 1996*), but contrasts with the chick where a 'transition zone' occurs between primary and secondary neurulation. The chick primary (dorsal) and secondary (ventral) neural tubes overlap for a short length of the spinal axis (*Dryden, 1980*) and cells ingressing at the node-streak border participate in a distinct 'junctional neurulation' process with essential function of the *Prickle-1* gene (*Dady et al., 2014*). Human 'junctional neurulation' has also been invoked as an explanation for an unusual form of spinal dysraphism where primary and secondary neural tubes are physically and functionally separated from each other, with no intervening neural tissue (*Eibach et al., 2017*). However, the lack of any evidence for 'junctional neurulation' in the present study of human embryos casts doubt on the proposed developmental origin of this rare dysraphic defect.

## Mode of development of the human secondary neural tube

Secondary neurulation in mouse and rat involves formation of a single 'rosette' structure caudally, in which cells aggregate ('condense') from the dorsal tailbud mesenchyme, with subsequent (more rostral) organisation of the cells around a single lumen. This process is driven by apical junction formation, not by cell death (*Butcher, 1929*; *Schoenwolf, 1984*; *Kostović-Knezević et al., 1991*; *Nievelstein et al., 1993*). In chick, by contrast, a caudal-to-rostral sequence of events occurs in which several independent lumens arise in the dorsal tailbud mesenchyme and, at more rostral levels, these coalesce to form the single lumen of the secondary neural tube (*Criley, 1969*; *Schoenwolf and Delongo, 1980*; *Yang et al., 2003*). Coalescence is a cell intercalation process driven by *TGFβ/SMAD3* signalling (*Gonzalez-Gobartt et al., 2021*).

The mode of formation of the human secondary neural tube is controversial. Multiple neural tube lumens have often been observed in the developing or regressing tail (*Bolli, 1966*; *Lemire, 1969*; *Hughes and Freeman, 1974*; *Fallon and Simandl, 1978*; *Saitsu et al., 2004*; *Pytel et al., 2007*; *Yang et al., 2014*), whereas other studies identify only a single lumen (*Müller and O'Rahilly, 1987*; *Nievelstein et al., 1993*). Here, we found multiple secondary neural tube lumens in 60% of human embryos. An important question is whether such multiple lumens are part of the normal secondary neurulation process in humans – thus making the human more similar to chick than mouse, as has been claimed (*Pang, 2020*). Alternatively, multiple lumens could arise through later 'splitting' of the previously formed neural tube. In mice, neural tube duplication is part of several mutant phenotypes and, when present, is a sign of pathology (*Cogliatti, 1986*).

A limitation of previous studies is the paucity of information on the morphology of the secondary neural tube at specific rostro-caudal axial levels. To shed light on this question, we examined human embryonic tails with the aim of determining the axial sequence of secondary neurulation events. Our findings show that multiple lumens, if present, feature at relatively rostral (mature) levels of the secondary neural tube and are absent from the most caudal (immature) levels, close to the tailbud. This applies to embryos throughout the secondary neurulation process (CS13–17), and argues strongly against coalescence of chick-like multiple lumens as a feature of normal human secondary neurulation. A recent review has drawn the same conclusion (*Catala, 2021*). Hence, 'splitting' of the human secondary neural tube appears a common but not obligatory phenomenon, perhaps reflecting changes related to tail regression.

## Tailgut: origin and mode of regression

The tailgut is an extension of the hindgut, beginning caudal to the level of the cloacal plate (future anus), which is located ventral to somite 29 in the mouse (*Nievelstein et al., 1993*). The tailgut forms and then regresses in both tailed (mouse, rat) and non-tailed (chick, human) animals. Interestingly, human tailgut loss has been described as involving rostral-to-caudal degeneration rather than a more intuitive caudal-to-rostral loss (*Kunitomo, 1918*; *Fallon and Simandl, 1978*). Consistent with this, the tailgut lumen persists longest at the tail tip in rat (*Butcher, 1929*; *Qi et al., 2000*) and mouse (*Nievelstein et al., 1993*). Our finding of a rostral-to-caudal diminution in tailgut size and cell number is consistent with tailgut loss at rostral before caudal levels also in human embryos.

In contrast to the consensus on tailgut regression, there is disagreement over the developmental origin of the tailgut. Anatomical and histological studies in rat, mouse, and human often conclude that the tailgut originates by mesenchyme-to-epithelium transition of tailbud cells, in a manner similar to the origin of the secondary neural tube (*Švajger et al., 1985*; *Gajović et al., 1989*; *Gajović et al., 1993*; *Nievelstein et al., 1993*). However, others consider the tailgut to arise by caudally directed extension of the hindgut (*Jolly and Ferester-Tadie, 1936*). Grafting of the E10.5 mouse tailbud beneath the kidney capsule produced no evidence of gut epithelial differentiation, in contrast to primitive streak/tailbud fragments at E8.5 and E9.5 which regularly produced this derivative. This finding is consistent with loss of gut-forming potential in the later stage tailbud (*Tam, 1984*).

The question of tailgut origin can also be considered in light of the identification of NMPs: the stem cell population for tissues of the caudal embryonic region (*Wilson et al., 2009*; *Wymeersch et al., 2021*). A retrospective clonal analysis found gut endoderm only as part of rostrally derived clones, unlike neural tube and paraxial mesoderm that were represented in clones extending into the tailbud at E10.5 (*Tzouanacou et al., 2009*). This led to the idea that NMPs are bipotential, forming neural and paraxial mesodermal derivatives, whereas the endodermal lineage is set aside separately, early in gastrulation. These findings are consistent with results of DiI-based lineage tracing and tissue grafting experiments (*Cambray and Wilson, 2002*; *Cambray and Wilson, 2007*) which show that the NMP population at the chordoneural hinge region of the tailbud is fated to form neural and mesodermal derivatives, but not tailgut. Further support for this concept comes from the finding of a proliferative zone at the hindgut tip, which is required to generate the colon by caudally directed gut extension (*Garriock et al., 2020*). It will be interesting to determine whether a similar mechanism underlies tailgut development.

## Mechanism of cessation of tail elongation

Termination of axial elongation is highly species-specific, occurring in embryos with fewer than 40 somites in human (*Table 2*), at ~52-somite stage in chick, and in embryos with 65 somites in rat and mouse (*Olivera-Martinez et al., 2012*). One question is whether the underlying molecular and cellular mechanisms are shared, despite these variations in timing, or are fundamentally different between species. Our findings with human embryos support a shared mechanism, as we find that expression of *FGF8* and *WNT3A* are developmentally regulated in close relationship to the time-course of axial elongation, similar to that in rodent embryos. Moreover, cessation of tail growth in the mouse has been linked to a burst of apoptosis in the tailbud around E13.5 (*Wilson et al., 2009*), and we detected an analogous burst of apoptosis in the CS15 human tailbud. Hence, a similar mechanism may underlie growth termination of the much shorter human embryonic tail.

## Type and timing of cell death during tail regression

While programmed cell death is recognised to participate in tail regression, the precise mode of cell death has been debated. In immunohistochemistry studies, it was concluded that apoptosis occurs only in the human cranial embryonic region, and non-apoptotic ('necrotic-like') death of tail structures was identified in the regressing human tail (*Sapunar et al., 2001*; *Vilović et al., 2006*). In contrast, cell death during chick tail regression was shown to involve caspase-dependent apoptosis (*Miller and Briglin, 1996*). Using anti-caspase 3 and TUNEL methods, we identified apoptosis in the human tail, with patterns of cell death occurring in a closely similar way between human and mouse embryos. We conclude that caspase-dependent apoptosis is the predominant mode of cell loss during tissue regression in the tails of both mouse and human.

## Conclusions

The findings of this study show a close parallel between human and rodent embryos in several features of low spinal development: completion of primary neurulation, transition to secondary neurulation, cellular mechanism of secondary neural tube formation, and molecular basis of cessation of tail elongation. In contrast, some aspects of chick low spinal development – often cited as an accurate model for human – are not represented in the human embryos, indicating the need for caution in extrapolating findings from birds and lower vertebrates to humans. The main differences between human and mouse/rat tail development relate to timing with, for example, formation of a new somite every 7 hr in humans, compared with 2 hr in mouse/rat, and termination of tail elongation at the 36- to 37-somite stage in human, compared with the 65-somite stage in tailed rodents. While tail regression occurs completely in human embryos, it is noteworthy that the tail of mouse/rat embryos also regresses partially, with loss of secondary neural tube and tailgut, despite maintenance of an overall tail structure. An intriguing observation is the presence of secondary neural tube splitting in apparently normal human embryos, whereas this is seen only under pathological conditions in rodents. Future work in human embryos and organoids may shed light on the mechanisms(s) of this phenomenon.

# Materials and methods

**Key resources table**

| Reagent type (species) or resource | Designation | Source or reference | Identifiers | Additional information |
|---|---|---|---|---|
| Strain, strain background (mouse) | CD1 | Charles River UK | Strain Code 022 | https://emodels.criver.com/en/page/species |
| Biological sample (human embryos) | Human embryos | MRC/Wellcome Human Developmental Biology Resource | NA | https://www.hdbr.org/ |
| Antibody | Rabbit polyclonal anti-cleaved caspase-3 (Asp 175) antibody | Cell Signalling | Cat. No. 9661 | Used at 1/1000 (wax sections) and 1/250 (cryosections) https://www.cellsignal.com/browse?categories=Primary%20Antibodies |
| Antibody | Donkey anti-Rabbit IgG (H+L) Highly Cross-Adsorbed Secondary Antibody, Alexa Fluor 647 | Thermo Fisher Scientific | Cat. No. A-31573 | Used at 1/250 https://www.thermofisher.com/antibody/product/Donkey-anti-Rabbit-IgG-H-L-Highly-Cross-Adsorbed-Secondary-Antibody-Polyclonal/A-31573 |
| Sequence-based reagent | Human *FGF8* DNA sequence | NIH National Library of Medicine | NM_033165.5 | https://www.ncbi.nlm.nih.gov/nuccore/NM_033165 |
| Sequence-based reagent | Human *WNT3A* DNA sequence | NIH National Library of Medicine | NM_033131.4 | https://www.ncbi.nlm.nih.gov/nuccore/NM_033131.4 |
| Commercial assay or kit | ApopTag Peroxidase In Situ Apoptosis Detection Kit | Sigma-Aldrich | Cat. No. S7100 | https://www.merckmillipore.com/GB/en/product/ApopTag-Peroxidase-In-Situ-Apoptosis-Detection-Kit,MM_NF-S7100 |
| Software, algorithm | Fiji software | ImageJ | Free downloads | https://imagej.net/software/fiji/downloads |
| Software, algorithm | AxioVision v4.8.2 software | Carl Zeiss | 410130-0600-000 | https://www.fishersci.pt/shop/products/axiovision-rel-4-8-2-software/11875113 |

## Human embryos

All embryos were obtained from the MRC/Wellcome Human Developmental Biology Resource (HDBR; https://www.hdbr.org/) with UK ethics committee approval and written consent of donors. Embryos were collected on ice in L-15 medium, rinsed in phosphate-buffered saline (PBS), and fixed overnight at 4°C in 4% paraformaldehyde (PFA) in PBS. Embryos were assigned to CS using morphological criteria (*O'Rahilly and Muller, 1987*; *Bullen and Wilson, 1997*) and to 2-day post-conception intervals for regression analysis based on timings in Table 0-1 of *O'Rahilly and Muller, 1987*. Only embryos that had normal external morphology and a normal karyotype were included in the study. Screening for aneuploidy was performed on all embryos, either by conventional karyotyping or by quantitative fluorescent polymerase chain reaction (*Badenas et al., 2010*).

## Mouse embryos

Mouse studies were conducted under auspices of the UK Animals (Scientific Procedures) Act 1986 and the National Centre for the 3Rs' *Responsibility in the Use of Animals for Medical Research* (2019). Random-bred CD1 embryos were collected from pregnant females between E10.5 and 14.5 (E0.5 is the day of finding a copulation plug). Embryos were dissected in Dulbecco's modified Eagle's medium, rinsed in PBS, and fixed in 4% PFA overnight.

## Embryo measurements

Measurements on human embryos were made post-fixation using an eyepiece graticule on a Zeiss SV6 stereomicroscope. Crown-rump length was measured as the maximum distance from the top of the head to the base of the spine. Tail length was measured along the ventral surface, from the tail tip to the point where the tail joined the trunk. The distance from the tail tip to the caudal edge of the caudal-most somite was also measured. Somites were counted in total or, where indistinct more rostrally, the total number was estimated by considering the somite immediately rostral to the hind-limb bud as somite 24. Analysis of PNP closure by somite stage (*Figure 1I and J*) was performed using archival embryo images, with PNP length measurements normalised to caudal somite length in the same embryo (both measured in pixels on photomicrographs).

## H&E histology

PFA-fixed caudal embryonic regions were dissected away from the remainder of the embryo and dehydrated through an ascending alcohol series to Histoclear (National Diagnostics), embedded in 56°C paraffin wax, and sectioned transversely at 7 µm thickness on a rotary microtome. For haematoxylin and eosin (H&E) staining, slides were dewaxed in Histoclear, and rehydrated through a descending alcohol series from 100% ethanol to water, then placed sequentially in: filtered Harris's haematoxylin (3 min); running tap water (1 min); Scott's Tap Water substitute (20 g sodium hydrogen carbonate+3.5 g magnesium sulphate in 1 l distilled water; ~3 s); running tap water (1 min); 95% ethanol (1 min); eosin (3 min); running tap water (1 min); 95% ethanol (1 min); 100% ethanol (2×1 min); Histoclear (2×5 min). Slides were mounted with DPX.

## Immunohistochemistry

Initial studies used wax sections (*Figure 3*), and subsequently cryosections were used (*Figure 4*). For the latter, PFA-fixed tissue was dehydrated to 30% sucrose in PBS and stored at 4°C. Tissues were incubated for ~6 hr in 30% sucrose+7.5% gelatine in PBS at 56°C, positioned in the gelatine mix at room temperature and allowed to set, and stored at –80°C. Gelatine-embedded samples were sectioned on a cryostat at 15 µm thickness with a sample temperature of –23°C and an ambient temperature of –25°C. For staining, wax sections were rehydrated as for H&E, then blocked and incubated with antibodies as below. Cryosections were incubated in PBS at 37°C to melt the gelatine, and antigen retrieval was performed using a decloaking chamber. Slides were incubated at 110°C for 2 min in 10 mM sodium citrate+0.05% Tween 20 in water (pH 6), and then returned to room temperature. Slides were rinsed in PBS+0.1% Triton (PBST) and blocked for 1 hr in PBST+0.15% glycine+10% sheep serum. Blocking solution was removed, followed by incubation overnight at 4°C in primary antibody solution: rabbit polyclonal anti-cleaved caspase-3 (Asp175; Cell Signalling, Cat. No. 9661) diluted in PBST+1% sheep serum at 1:1000 for wax sections or 1:250 for cryosections. The next day, slides were washed 3× for 5 min each in PBST, then incubated in a humidity chamber for 1 hr in secondary antibody solution: 1:250 Alexa Fluor donkey anti-rabbit 647 (Thermo Fisher Scientific, Cat. No. A-31573). Slides were washed 3× for 5 min in PBST, counterstained for 5 min in 1:5000 DAPI, washed a final 2×5 min in PBS, and mounted using ProLong Gold Mountant (Thermo Fisher Scientific, Cat. No. P36930).

## TUNEL staining

TUNEL (Apoptag; Sigma-Aldrich) was performed according to the manufacturer's instructions. Sections were counterstained with methyl green (Vector Labs, H-3402).

## Morphometric analysis of embryo sections

All image analysis was carried out using Fiji Is Just ImageJ (FIJI) software (*Schindelin et al., 2012*). Area was calculated using the polygon tool. Nuclei were counted using the multi-point tool. Neural

tube, notochord, and tailgut areas, and tailgut nuclear number, were plotted against total area of the tail section, as a measure of position along the rostro-caudal axis.

## Whole-mount in situ hybridisation

Digoxigenin (DIG)-labelled mRNA probes for human *FGF8* (reference sequence NM_033165.5) and *WNT3A* (reference sequence NM_033131.4) were designed for in situ hybridisation. Human embryos or isolated caudal regions were fixed in 10% formalin, washed in PBS with 0.1% Tween (PBT), and processed for whole-mount in situ hybridisation. Samples were bleached in 6% hydrogen peroxide, digested in a 5 µg/ml proteinase K-PBT solution, followed by a wash in 2 mg/ml glycine, and subsequently fixed in 0.2% glutaraldehyde made up in 4% PFA. Samples were then incubated in pre-hybridisation mix (50% formamide, 1% sodium dodecyl sulfate [SDS], 5× saline sodium citrate [SSC], 50 µg/ml yeast tRNA, and 50 µg/ml heparin), and hybridised with the corresponding DIG-labelled mRNA probes overnight at 70°C. Hybridised probes were fixed using fixative wash solutions (solution 1: 50% formamide, 5× SSC, and 1% SDS at 70°C; and solution 2: 50% formamide, 2× SSC, and 1% SDS at 65°C). The samples were then blocked in 10% heat inactivated sheep serum, and incubated with anti-DIG-alkaline phosphatase antibody (Roche) solution. Development of the colour signal was carried out in nitrotetrazolium blue and 5-bromo-4-chloro-3-indole solution. Whole-mount images were taken using a DFC490 camera (Leica) connected to a Stemi SV11 stereo-microscope (Zeiss), and then embedded in gelatin-albumin for vibratome sectioning at a thickness of 40 µm. Sections were imaged using AxioVision v4.8.2 software on an Axioplan 2 microscope (Zeiss).

## Statistical analysis

Linear regression analysis (*Figure 1*) and one-way analysis of variance on ranks (*Figure 2*) were performed using Sigmaplot v14.5.

## Systematic literature review (*Supplementary file 1*)

PubMed (https://pubmed.ncbi.nlm.nih.gov/) was searched for a variety of term combinations that included: human, embryo, embryonic development, neuropore, primary neurulation, secondary neurulation, neural tube, organogenesis, tail, transcrptomics, gene expression. Retrieved papers were scanned for relevance sequentially using the title, abstract, and full text, with non-qualifying papers dismissed at each stage. Additional relevant papers were identified from the bibliographies of the retrieved papers. Qualifying papers (n=28) were those that presented original data on human low spinal/tail development, using embryo samples not described in other studies.

## Acknowledgements

This work was supported in part by the Wellcome Human Developmental Biology Initiative (HDBI: grant 215116/Z/18/Z). Human embryonic material was provided by the MRC/Wellcome Human Developmental Biology Resource (grant MR/R006237/1; https://www.hdbr.org).

---

## Additional information

### Funding

| Funder | Grant reference number | Author |
| --- | --- | --- |
| Wellcome Trust | 10.35802/215116 | Andrew J Copp |
| Medical Research Council | MR/R006237/1 | Andrew J Copp |

The funders had no role in study design, data collection and interpretation, or the decision to submit the work for publication. For the purpose of Open Access, the authors have applied a CC BY public copyright license to any Author Accepted Manuscript version arising from this submission.

## Author contributions
Chloe Santos, Abigail R Marshall, Gabriel L Galea, Formal analysis, Investigation, Writing – review and editing; Ailish Murray, Priyanka Narayan, Sandra CP de Castro, Eirini Maniou, Investigation, Writing – review and editing; Kate Metcalfe, Data curation, Formal analysis, Investigation, Writing – review and editing; Nicholas DE Greene, Supervision, Writing – review and editing; Andrew J Copp, Conceptualization, Data curation, Formal analysis, Supervision, Funding acquisition, Investigation, Writing - original draft, Project administration, Writing – review and editing

## Author ORCIDs
Nicholas DE Greene ![orcid] https://orcid.org/0000-0002-4170-5248
Gabriel L Galea ![orcid] https://orcid.org/0000-0003-2515-1342
Andrew J Copp ![orcid] https://orcid.org/0000-0002-2544-9117

## Ethics
Human subjects: All human embryos were obtained from the MRC/Wellcome Human Developmental Biology Resource (HDBR; https://www.hdbr.org/) with UK ethics committee approval and written consent of donors.

Mouse studies were conducted under auspices of the UK Animals (Scientific Procedures) Act 1986 and the National Centre for the 3Rs' Responsibility in the Use of Animals for Medical Research (2019).

Reviewer #1 (Public review): https://doi.org/10.7554/eLife.88584.3.sa1
Reviewer #2 (Public review): https://doi.org/10.7554/eLife.88584.3.sa2
Author response https://doi.org/10.7554/eLife.88584.3.sa3

---

# Additional files

## Supplementary files
• Supplementary file 1. Table of publications from a review of the literature on human secondary neural tube and body formation. Rows show individual publications; columns show the topics covered in publications.

• Supplementary file 2. Source data on human embryos as summarised in *Figure 2I–K* and *Table 2*. Each line in the table corresponds to a different human embryo (n=37). Measurements of crown-rump length, tail length, and tail length distal to somites were not available for all embryos. Data in *Figure 2I–K* and *Table 2* are based on the available data.

• MDAR checklist

## Data availability
All data generated or analysed during this study are included in the manuscript and supporting files.

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
