## [Editor Report · eLife Assessment]

This is a **fundamental** study into human spinal neurulation, which substantially advances our understanding of human neural tube closure. Crucial unanswered questions in the field currently rely on model systems, not faithful to human development. The evidence provided is **compelling**, with a large number of specimens and the rigorous use of state-of-the-art methodology providing robustness. The work will be of broad interest to developmental biologists, embryologists, and medical professionals working on neural tube defects, and will act as a precious reference resource for future studies.

---

## [Referee Report · Reviewer #1 (Public review)]

Summary:

The authors analyzed 108 human embryos in order to address outstanding questions about human lower spinal development and secondary neural tube formation. Through whole embryo imaging and histologic analysis, they have provided exceptional quantification of the timing of posterior neuropore closure, rate of lower spinal somite formation, and formation and regression of the human tail. Their analysis has also provided convincing qualitative evidence of the cellular and molecular mechanisms at play during lower spinal development, by identifying the presence of caspase-dependent programmed cell death and the dynamic expression of FGF8/WNT3A within the elongating embryo. Interestingly, they identified multiple polarized lumens within the site of secondary neural tube formation, and added a solid argument for the mode of formation of this structure; however, the evidence for a conclusive morphogenetic mechanism remains elusive. Finally, the authors provided a substantial review of the existing publications related to human lower spinal development, creating an excellent reference and demonstrating the importance of continuing to use each of these precious samples to further advance our understanding of human development.

Strengths:

This manuscript provides an excellent window into the key morphogenetic events of human caudal neural tube formation. Figures 1 and 2 provide beautiful images and quantification of the developmental events, enabling comparison to models that are currently in use, including model organisms and the developing spinal organoid field. The characterization of somite development and later regression is particularly important.

In Figures 3 and 4, the authors use immunohistochemistry to examine the cellular death mechanisms and spatiotemporal organization of tissue regression within the tail. They demonstrate a proximal to distal tapering of the overall tail and neural tube areas that is not present for the notochord and reveal a proximal to distal degeneration of the tailgut, similar to what is observed in rodents. The identification of caspase-dependent cell death within the human tail provides an explanation for the mechanism of this regression, especially given the notable lack of presence of any gross necrosis.

Next, the authors have addressed current questions regarding the molecular pathways present during elongation of the embryo and later regression of the tail structure. The in situ hybridization experiments in Figure 5 show important evidence for a maintained neuromesodermal progenitor pool of stem cells that promote axial elongation. Additionally, the authors have conducted serial transverse sections of the tail to better understand the formation of the secondary neural tube in humans. They found a rodent-like formation involving a singular rosette caudally at the tailbud tip, and that multiple lumens, if present, were located more rostrally. This clearly differs from chick secondary neurulation. Finally, as mentioned above, the non-trivial collection and review of the existing human secondary neural tube and body formation literature is an important tool that organizes and synthesizes ~ 100 years of observations from precious human samples.

Weaknesses:

(1) The non-pathologic presence of multiple polarizations in human tails compared to the rodent pathogenic counterpart is interesting given that rodents obviously maintain this appendage that is lost in humans. A clear mechanism for how the secondary tube becomes continuous with the primary tube and how this relates to the presence of multiple polarizations in humans remains elusive.

---

## [Referee Report · Reviewer #2 (Public review)]

This study utilizes an extensive series of neurulation human embryos to address several open questions about the similarities and differences between human primary and secondary neurulation in the tail. Results are compared to other model systems, such as the chicken and rodent. Histology, in situ hybridization, and apoptosis analysis provide molecular data about how the tail regresses in the human embryo. The number of embryos utilized for the analysis and the quality of the histological analysis provide robustness to the findings.

Comments on revised version:

The authors have meticulously addressed all the concerns raised by the reviewers, using new data and modifications to the text to further strengthen the quality of the manuscript.

This is a fabulous manuscript. I have nothing more scientifically to critique.

---

## [Author Response]

The following is the authors’ response to the original reviews.

**Reviewer #1 (Public Review)**
(1) The identification of the proximal to distal degeneration of the tailgut within the human tail is difficult to distinguish with the current images present in Figure 3. A picture within a picture of the area containing the tail gut could be provided to prominently demonstrate the cellular architecture. Additionally, quantification of the localization of apoptosis would strongly support this observation, as well as provide a visualization of the tail's regression overall. For example, a graph plotting the number of apoptotic cells versus the rostral to caudal locations of the transverse sections while accounting for the CS stage of each analyzed embryo could be created; this could even be further broken down by region of tail, for example, tailgut, ventral ectodermal ridge, somite, etc.

To provide more information on apoptosis, we prepared serial sections from an additional 6 human tails, 5 of which were processed for fluorescence anti-caspase 3 immunohistochemistry with DAPI staining (Fig 4) and H&E (Fig 6). This confirmed our previous finding of apoptosis especially in the tailgut and ventral mesoderm. We have not quantified the apoptosis, given the difficulty of deciding whether anti-caspase signals represent single or multiple dying cells. Instead, we performed a tissue area analysis from caudal to rostral along the tail (new section on p 9). This shows a progressive enlargement of the neural tube, no change in the notochord and a striking reduction in tailgut area (Fig 4C,D). The smaller tailgut has fewer nuclei in cross section rostrally compared with more caudally (Fig 4E). Given that apoptosis is present in the tailgut at all rostro-caudal levels, this is consistent with a rostralto-caudal loss of the tailgut, as is also found in mouse and rat embryos.

(2) The identification of the mode of formation of the secondary neural tube is probably the most interesting question to be addressed, however, Figure 7's evidence is not completely satisfying in its current form. While I agree that it is unlikely that multiple polarization foci form within the most caudal part of the tail and coalesce more rostrally, I am equally unsure that a single polarization would form rostrally and then split and re-coalesce as it moves caudally, as is currently depicted by 7B. Multiple groups have recently shown the influence of geometric confinement on neuroectoderm and its ability to polarize and form a singular central lumen (Karzbrun 2021, Knight 2018), or the inverse situation of a lack of confinement resulting in the presence of multiple lumens. The tapering of the diameter of the tail and its shared perimeter and curvature with the polarization bears a striking resemblance to this controlled confinement. An interesting quantification to depict would include the number of lumens versus the transverse section diameter and CS stage to see if there is any correlation between embryo size and the number of multiple polarizations. Anecdotally, the fusion of multiple polarizations/lumens tends to occur often in these human organoid-type platforms, while splitting to multiple lumens as the tissues mature does not. Other supplements to Figure 7 could include 3D renderings of lumens of interest as depicted in Catala 2021, especially if it demonstrates the recoalescence as seen in 7B. The non-pathologic presence of multiple polarizations in human tails compared to the rodent pathogenic counterpart is interesting given that rodents obviously maintain this appendage while it is lost in humans.

The additional 6 sectioned human embryo tails (as described above) provide further information in support of the original findings of the paper: (i) that the secondary neural tube formation initially involves a single lumen, and (ii) that neural tube duplication occurs in many tails at more rostral levels. Neural tube duplication was observed in 15/25 of our sectioned tails: hence, overall 60% of human tails exhibited neural tube duplication in this study. We have replaced all the cross sectional images in the original Fig 7 (now Fig 6) to better illustrate the findings of neural tube duplication at relatively rostral levels of the human tail. Additionally, the axial position of sections containing duplicated neural tube are indicated by arrows in the graph of neural tube areas (Fig 4C). From this analysis it appears that neural tube duplication is not contingent on an increasing tail diameter, as raised by the reviewer, because some tails show a transition to neural tube duplication, and then return to an single lumen morphology more rostrally. While the 3D renderings of lumens would be interesting, we consider it beyond the scope of the present study.

(3) Of potential interest is the process of junctional neurulation describing the mechanistic joining of the primary and secondary neural tube, which has recently been explored in chick embryos and demonstrated to have relevance to human disease (Dady 2014, Eibach 2017, Kim 2021). While it is clear this paper's goal does not center on the relationship between primary and secondary neurulation, such a mechanism may be relevant to the authors' interpretation of their observations of lumen coalescence. I wonder if the embryos studied provide any evidence to support junctional neurulation.

We agree this is an important point to address in the paper, and a new section has been inserted in the Discussion: ‘Transition from primary to secondary neurulation’ (pp 13-14). In brief, we find no evidence for a specific mode of ‘junctional neurulation’ in the human embryos. In any event, its existence is hypothetical in humans, suggested largely as an ‘embryological explanation’ for the finding of rare interrupted spinal cord defects in neurosurgical patients (Eibach, 2017). In chick neurulation there is longitudinal dorso-ventral overlap between the primary and secondary neural tubes (Dryden, 1980), with the junctional zone derived from ingressing cells at the node-streak border (Dady, 2014), a known source of neuromesodermal progenitors (NMPs). However, this is a very different developmental situation from the human so-called ‘junctional neurulation’ defect (Eibach, 2017), in which the spinal cord is physically and functionally interrupted, with only a rudimentary filament connecting the rostral and caudal parts.

**Reviewer #1 (Recommendations For The Authors):**
(1) Figures 3, 4, and 7, would be easier to digest quickly with inclusions of labels that mark the rostral and caudal transverse sections. For example, "caudal" over 3G and "rostral" over 3F.

Figures 3 and 4 have been combined to form revised Figure 3, and the rostral/caudal sections are no longer included, as these are superseded by the new Figure 4. Similarly Figure 7 has been replaced by new images in the revised Figure 6, with clear labelling of axial levels.

(2) The manuscript does a nice job of comparing and contrasting the human findings to mouse, however, there are several instances where it would be nice to continue this trend within the text, such as including the rate of somite formation for rodents in the sections that you state the quantified human and published organoid findings, as well as the total number of somite rodents' exhibit. Additionally, the last sentence of the "Morphology of human PNP closure" section correctly states that human PNP's seem to close via Mode 2 neurulation that is seen in the mouse. However, my read of the literature (published by Dr. Copp) demonstrates that the PNP in mice actually closes via Mode 3 at the most caudal portion. If this is the case, it would be pointed to explicitly state that regionally dependent morphogenetic difference between the two species.

We agree these are important points to include. The additional somite data (for mouse) has been inserted in the Results section on ‘Somite formation’ (p 8), and the apparent absence of Mode 3 during human spinal neural tube closure is now included in the new Discussion section, ‘Transition from primary to secondary neurulation’ (pp 13-14).

(3) The introduction to secondary neural tube formation with the hypothesis diagrams in Figure 7 is slightly jarring. At the beginning of the Figure, a schematic depicting the morphogenetic differences between primary and secondary would be helpful in introducing the readership to these complex embryologic events. An example of this could be similar to Figure 1 in Dr. Copp's paper:

Nikolopoulou, E., et al. Neural tube closure: cellular, molecular and biomechanical mechanisms. Development 144, 552-566 (2017).

We feel that a summary diagram of primary and secondary neurulation would simply reproduce diagrams that are already widespread in the literature. As noted by the Reviewer, our article in Development (Nikolopoulou, 2017) contains just such a summary diagram as Figure 1. Therefore, we prefer to explicitly cite this article/figure in our Introduction (see modified first sentence, third paragraph, p 3), so that readers can consult the freely accessible Nikolopoulou review for more detail. The diagram in Figure 7 (now revised Figure 6) has been completely redrawn to make much clearer the hypotheses being examined in the study of human secondary neural tube formation, and neural tube duplication.

(4) Finally, a matter of semantics, the second paragraph of the introduction describes myelomeningocele as a neurodegenerative defect, while it is true amniotic fluid further degrades exposed neural tissue while exposed, to me, the term neurodegenerative defect suggests a lifelong degeneration, which is not the case for human patients. Perhaps shortening to neurological defect is a compromise. Thank you for the important and interesting work.

We agree that ‘neurodegenerative’ can mean different things to different people. Literally, it refers to degeneration of neural tissue, which of course includes neuroepithelial loss due to amniotic fluid action in the uterus. Nevertheless, to avoid confusion, the word has been removed and the sentence expanded to include a reference to the adverse effects of amniotic fluid on the exposed neuroepithelium (see Introduction, second paragraph, p 3).

**Reviewer #2 (Public Review)**
It is not clear how the gestational age of the specimens was determined or how that can be known with certainty. There is no information given in the methods on this. With this in mind, bunching the samples at 2-day intervals in Figure 1J will lead to inaccuracies in assessing the rate of somite formation. This is pointed out as a major difference between specimens and organoids in the abstract but a similar result in the results section. The data supporting either of these statements is not convincing.

Human embryos were assigned to Carnegie stages based on standard morphological criteria. This was stated, with references, in the first Results paragraph, and we have now also included this information in the Methods (first paragraph, p 19). We assigned the embryos to 2-day intervals based on the standard literature timing of these Carnegies stages, as described in O’Rahilly and Muller (1987). We have clarified both Carnegie staging and assignment of embryos to 2-day intervals in a new sentence within the Methods, first paragraph, p 19. “Embryos were assigned to Carnegie Stages (CS) using morphological criteria (O'Rahilly and Muller 1987; Bullen and Wilson 1997) and to 2-day post-conception intervals for regression analysis based on timings in Table 0-1 of O’Rahilly and Muller (1987).” This has also been inserted in the legend to Figure 1J.

The regression analysis of somite number against days post-conception (Figure 1J) allowed a conclusion to be drawn on the rate of somite formation in early human embryos. We have added 95% confidence intervals to our finding of a new somite formed every 7.1 h in humans. We consider this to be important for comparison with non-human species and organoid systems. On p 8, second paragraph, we simply state our finding of a 7.1 h somite periodicity in human embryos, compared with 5 h in the organoid system (and 2 h in mouse and rat – as suggested by Reviewer 1). We are careful not to say it is a ‘major difference’ or ‘similar result’ in different parts of the paper, as the Reviewer has drawn attention to.Whenever possible, give the numbers of specimens that had the described findings. For example, in Figure 2C - how many embryos were examined with the massive rounded end at CS13? Apoptosis in Figures 3 and 4?

Numbers of embryos analysed in Figures 2 and 3 (the latter now a combined version of the original Figures 3 and 4) are shown in Table 2. We have also created a new Supplementary Figure 1 to show additional examples of human embryonic tails, which illustrate the consistency of morphology through the stages from CS13 to CS18. Numbers of samples that contributed to Figures 4-6 are detailed in the legends.

For Figure 2I-K, it would be informative to superimpose the individual data points on the box plots distinguishing males from females, as in Figure 1I.

This was attempted but the data points overlie the box plots and look confusing. Instead, we have created Supplementary Table 2 which gives the raw data on which Figure 2I-K are

based. We have also drawn attention to the fact that not all embryos yielded all types of measurement, especially tail lengths.

Is it possible to quantitate apoptosis and proliferation data?

We have not quantified apoptosis, given the difficulty of deciding whether anti-caspase signals represent single or multiple dying cells. Instead, we performed a new tissue area analysis along the body axis, which has shed light on the possible direction (rostral to caudal) of tailgut loss in the human caudal region (see response to Reviewer 1 above). Since the cell proliferation data were limited in extent, and not a major focus of the paper, we have removed that analysis completely from the revised version.

The Tunel staining in Figure 3 is difficult to make out.

We have extended our analysis of anti-caspase 3 immunohistochemisty and removed the TUNEL images.

**Reviewer #2 (Recommendations For The Authors)**
The anatomy of the sections in Figures 3, 4, and 7 is difficult to discern. Is it possible to insert adjacent panels tracing and labeling the structures in each panel? Also, drawings showing the axial level of each section would be helpful.

To clarify the axial levels of sections, we have inserted images of mouse and human embryos as parts A and B of the revised Figure 3. We have tried to clarify the morphology of sections by labelling all relevant structures in the sections themselves.

High-magnification views of the tailbud in Figure 5 would be more informative. Staining is difficult to see after CS13. The low-magnification views can be shown in an insert. Figures 5 and 6 can be combined.

At the reviewer’s suggestion, we have merged Figures 5 and 6 into a revised Figure 5. Now, the sections provide higher magnification images of the areas of expression as shown in the lower magnification whole mount images. We feel this makes the gene expression findings much clearer than before.

Some of the writing in the abstract, introduction, and results is very descriptive, with a lack of summary and integration of information. For instance, the abstract could be rewritten to include an overall conclusion at the end and a better description of the longstanding questions addressed. Moreover, the abstract suggests multiple lumens are not found in human specimens. Another example is the second paragraph of the introduction lists various NTDs but doesn't provide an integrative conclusion of the information. The discussion is much better but lacks a conclusion at the end.

We agree that more concluding sentences should be used, as the Reviewer suggests. To this end, we have rewritten the Abstract (p 2) to emphasise the long-standing questions that our study addresses, and concluding sentences are now included in other places (e.g. somite results, p 8). A new ‘Conclusions’ section has been added at the end of the Discussion (pp 17-18).

ADDITIONAL CHANGES MADE TO REVISED MANUSCRIPT

Title. This has been amended to: “Spinal neural tube formation and tail development in human embryos” to reflect the greater focus on developmental events, and less on tail regression.

Additional studies have been added to Supplementary Table 1, to include the main transcriptomic studies of human embryos in the primary/secondary neurulation stage range. This takes the number of previous studies to 28 and the total number of embryos to 925. See p 4, top and p 12, first paragraph for corresponding changes to the text.

We added a sentence to the Discussion (p 13, first paragraph) to counter the claim that humans have undergone ‘tail-loss’, as included in Xia et al, 2024, “On the genetic basis of tail-loss evolution in humans and apes”. Nature 626:1042-8. Clearly, the human embryo is tailed, which undermines these authors’ statement.